# MECHANISTIC INTERPRETABILITY OF LLMS THROUGH NETWORK SCIENCE

## ABSTRACT

Understanding scaling laws and emergent abilities in Large Language Models (LLMs) remains a key challenge for interpretability. While much prior work in mechanistic interpretability has focused on learned representations, the attention matrix—which governs information flow—has received much less *attention*. Furthermore, the analysis of the attention matrix from a theoretical network science perspective has also not been done. In this work, we present a pipeline for dynamic graph construction from attention matrices, introduce a novel head aggregation technique based on entropy, and analyse the attention graphs from a network science perspective to draw interpretability insights. Our experiments show that the entropy-based head aggregation preserves attention details, and that key graph metrics—specifically the clustering coefficient and maximum pagerank—correlate with improved model correctness and emergent abilities in LLMs. Notably, our findings indicate that larger models exhibit higher maximum pagerank and lower clustering coefficients, suggesting they reason differently by attending more globally and selectively focusing on key hotspots.

## 1 INTRODUCTION

LLMs have demonstrated remarkable capabilities in a wide range of tasks, from natural language understanding to translation to problem-solving (Liu et al., 2024; Grattafiori et al., 2024; Team et al., 2024). However, reliable methods to interpret them are still under investigation. Furthermore, as these models scale in size, they exhibit emergent abilities—capabilities that were absent in smaller models but manifest at larger scales—and understanding the underlying mechanisms that give rise to such behaviours remains an open research challenge in interpretability. Most existing research has focused on understanding the representations formed within the model, such as how embeddings encode linguistic and conceptual information. In contrast, the structure of the attention mechanism, which governs information flow, has received comparatively less attention (Clark et al., 2019).

Transformers (Vaswani et al., 2023), the foundation of modern LLMs, operate on fully connected graphs where attention weights define how tokens interact. Furthermore, it is known that these graphs are not optimal (Zaheer et al., 2020), and graph rewiring should be applied. Therefore, this motivates a natural graph-theoretic approach to interpretability. If attention determines how information propagates through a model, then analysing attention patterns and their corresponding graph topologies may reveal differences in information processing across scales, which may explain scaling laws and emergent properties.

In this paper, we propose a new interpretability framework based on **Autoregressive Attention Graphs**, dynamic directed graphs constructed from attention matrices across transformer heads, layers, and timesteps. By applying network science techniques and computing relevant metrics, we analyse emergent properties in LLMs and characterize the differences in information flow between small and large models.

We hypothesize that the **Autoregressive Attention Graphs** of larger models may exhibit finer-grained structure and interpretable properties compared to the smaller models, which may derive from their better understanding of semantic, lexical, and structural relationships between words and concepts. Specifically, we explore whether larger models demonstrate more consistent attention patterns, more well-defined communities, and greater efficiency at information propagation. By

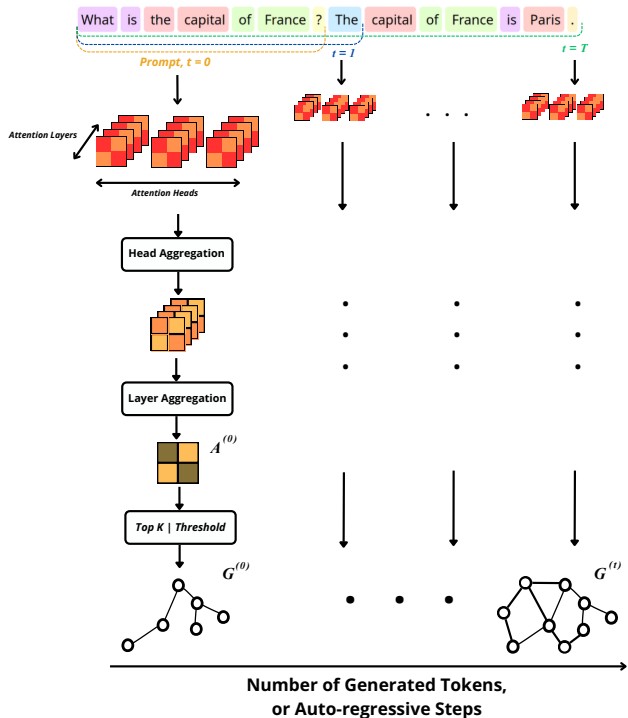

Figure 1: A visualization of the overall pipeline. For each autoregressive step of LLM token generation, we first aggregate individual attention heads through averaging or the entropy method we propose. Then, the resulting matrices are aggregated across layers using Attention Rollout (Abnar & Zuidema, 2020). Lastly, each of these aggregated matrices $A^{(t)}$ is transformed into a graph $G^{(t)}$ by either selecting the top k edges for each node or applying an edge weight threshold. These graphs form our dynamic graph, and we analyse the graph features for each of the static graphs, i.e., the graph corresponding to each timestep of token generation.

quantitatively comparing these graph-theoretic properties across model scales, we aim to provide novel insights into the mechanisms that drive emergent abilities in LLMs.

**Contributions:**

- We introduce **Autoregressive Attention Graphs**, a pipeline for constructing dynamic attention graphs, enabling a graph-theoretic analysis of attention in LLMs.

- We propose an entropy-based head aggregation method, and demonstrate that it preserves high-frequency attention details better than simple averaging.

- We analyse the **Autoregressive Attention Graphs** of LLMs across multiple tasks and models to uncover correlations with emergent properties.

**Key findings:**

- Graph metrics computed on the **Autoregressive Attention Graphs**, such as the clustering coefficient and maximum pagerank, show statistically significant correlations with model correctness.

- Larger models exhibit emergent attention patterns, with lower clustering coefficients and higher maximum pagerank values. This suggests that larger models reason differently, shifting their attention towards a more global regime with fewer key hotspots.

- Our network science approach provides a novel quantitative framework for linking attention dynamics to emergent capabilities in LLMs.

## 2 RELATED WORK

### 2.1 MECHANISTIC INTERPRETABILITY IN LLMS

Mechanistic interpretability aims to reverse-engineer the internal computations of large language models by mapping their computation to understandable algorithms or circuits. Early work by Olah et al. (2020) introduced the notion of *circuits*—coherent sub-graphs of neurons or attention heads that implement specific functions. Given that transformers route information explicitly through attention mechanisms, a significant body of work has focused on understanding attention head behaviour. For example, Clark et al. (2019) and subsequent studies have shown that certain heads consistently capture linguistic phenomena such as syntactic dependencies, co-reference, or positional relations. In machine translation models, Voita et al. (2019) demonstrated that a small subset of attention heads plays a critical role, while many appear redundant. Yet, little work has systematically examined attention structures across model scales using formal graph metrics.

### 2.2 EMERGENT PROPERTIES IN LLMS

Emergent abilities—capabilities that only manifest when models reach a certain scale—have attracted considerable attention in recent years. Wei et al. (2022) coined the term "emergent abilities" to describe phenomena that cannot be linearly extrapolated from smaller models, such as improved in-context learning or reasoning skills. While the scaling laws of LLMs predict general improvements in performance, the mechanisms responsible for this change in capabilities remain poorly understood (Schaeffer et al., 2023). Recent research has suggested that emergent abilities may arise either from the development of new internal circuits and information processing strategies (Olsson et al., 2022), or the wider diversity of data that larger models are trained on (Lu et al., 2024). These studies suggest that emergent properties are tied to both the architecture and the dynamics of attention. However, a systematic, graph-theoretic approach has yet to be explored.

### 2.3 INTEGRATING GRAPH-THEORETIC APPROACHES

Although network science has long been applied in other domains, its application to LLM interpretability remains nascent (Barbero et al., 2024; de Luca et al., 2024). Recent efforts, such as those by El et al. (2025), have begun to frame attention matrices as graphs, paving the way for systematic analyses using classical graph metrics. Our work builds on these insights by proposing *Autoregressive Attention Graphs*—dynamic, temporally evolving graphs that capture token-to-token attention during autoregressive generation. By relating graph metrics (such as clustering coefficients and pagerank) to model performance, we provide a novel quantitative framework for understanding the mechanisms underlying emergent behaviour.

## 3 CONSTRUCTION OF AUTOREGRESSIVE ATTENTION GRAPHS

### 3.1 ATTENTION AS A GRAPH STRUCTURE

During inference, the evolving attention of a Transformer can be seen as a dynamic, temporally evolving graph, where attention updates progressively modify token representations. This Attention Graph dynamically determines how different input tokens influence one another and evolves not only across layers but also across time steps during token generation.

Our formalization of an **Autoregressive Attention Graph** allows us to aggregate these matrices in a principled manner to effectively analyse the generation process.

The **Autoregressive Attention Graph** is a dynamic directed graph in which the timesteps represent autoregressive generation steps. Each new token generated by the model represents a new time step in the sequence, meaning that the structure of the **Autoregressive Attention Graph** is updated with every token in the output. The entire generation process can thus be viewed as a temporally evolving dynamic graph, where the model's attention shifts with each new token. For example, if a model is prompted to count to three and responds sequentially with the pseudo-tokens "token1, token2, token3" this constitutes a three-timestep series, where each timestep corresponds to the addition of a new node to the graph, and therefore the relative change of previous attention edges between prior

tokens and connections to the new one. Therefore, unlike static attention matrices, which capture the attentions of a single head and layer, **Autoregressive Attention Graphs** are constructed from a principled approach to allow us to study how information flow develops over time–or autoregressive generation timesteps.

By enabling a graph-theoretic analysis of attention, we can analyse the differences in how attention is distributed, such as the presence of highly connected hubs, hierarchical clustering, or long-range dependencies, and how these metrics may distinguish larger, more capable models from their smaller counterparts. By analysing these structures, we aim to uncover systematic trends that align with the emergence of new abilities.

This process comprises three distinct steps: head aggregation, layer aggregation, and graph construction. The overall pipeline described in this section can be seen in Figure 1.

## 3.2 HEAD AGGREGATION

Let us consider a fixed timestep $t$. For this timestep, $A_\ell^{(h)} \in \mathbb{R}^{n \times n}$ denotes the attention matrix at layer $\ell$ and head $h$, where $n$ is the sequence length or number of tokens seen thus far, $L$ is the number of layers, and $H$ is the number of heads per layer. We consider two methods for head aggregation:

### 3.2.1 AVERAGE HEAD AGGREGATION

A straightforward approach is to compute the aggregated attention matrix for each layer by averaging over all heads:

$$A_\ell = \frac{1}{H} \sum_{h=1}^{H} A_\ell^{(h)}.$$

However, as discussed in Section 5.2, this averaging process can obscure the unique contributions of individual heads and result in the loss of details. Specifically, it is known that attention heads attend differently to specific parts of the input. For some tokens, certain heads might contribute little to no attention while others contribute significantly; averaging these values may effectively cancel out these distinct details, appearing as if that token had been moderately attended to.

### 3.2.2 ENTROPY-BASED HEAD AGGREGATION

Similar to (Ali et al., 2025), this approach aggregates the attention heads by leveraging the entropy of each head's attention distribution and selecting a row from each head based on its entropy. For each token $i$ (i.e., for each row of the attention matrix), we compute the entropy of the attention distribution for that token at every head $h$:

$$E_{h,i} = -\sum_{j=1}^{n} \left(A_\ell^{(h)}\right)_{ij} \log \left(A_\ell^{(h)}\right)_{ij}.$$

For a given token $i$, let $\{E_{1,i}, E_{2,i}, \ldots, E_{H,i}\}$ denote the collection of entropies across heads. We then choose a head based on a quantile parameter $\alpha \in [0, 1]$:

- $\alpha = 0.0$ selects the head with the lowest entropy,
- $\alpha = 1.0$ selects the head with the highest entropy,
- $\alpha = 0.5$ selects the head corresponding to the median entropy.

If we denote by entropy_sorted_rows the indices of the sorted rows in ascending order by entropy value, then the selected head for token $i$ is given by:

$$h^*(i) = \text{entropy\_sorted\_rows}\Big[\lfloor \alpha \times (H-1) \rfloor\Big],$$

The aggregated attention matrix $A_\ell$ is then defined row-wise by selecting, for each token $i$, the attention distribution from the chosen head:

$$\forall i \in \{1, \ldots, n\}, \quad (A_\ell)_i = h^*(i).$$

This method allows us to control whether we wish to favour heads with higher entropy (i.e., those with more uniform attention distributions) or lower entropy (i.e., those with more concentrated attention), simply by adjusting the parameter $\alpha$.

## 3.3 Layer Aggregation

Given the head-wise aggregated matrix $A_\ell$ for each layer $\ell$, we then want to aggregate them across layers to have a final single graph per autoregressive step. We use the attention rollout technique from Abnar & Zuidema (2020), which can be summarized as:

$$\tilde{A}_\ell = \big((1 - \beta)A_\ell + \beta I\big)\tilde{A}_{\ell-1}$$

The $\beta$ parameter is used as a self-loop-inducing factor to prevent the lower-triangular matrix from collapsing the attention towards the first token when aggregating across layers.

Therefore, the final aggregated attention matrix across layers $\ell = 1, \ldots, L$ for time-step $t$ is:

$$\tilde{A}^{(t)} = \tilde{A}_L^{(t)} = A_L^{(t)} A_{L-1}^{(t)} \cdots A_1^{(t)}.$$

## 3.4 Dynamic Graph Construction

After computing the aggregated attention $\tilde{A}^{(t)}$ for each autoregressive timestep $t$, we remove the first token (to avoid the attention sink phenomena (Xiao et al., 2024)) and perform row-wise normalization.

Then, for each timestep $t$, we obtain an attention graph $G^{(t)}$ by applying either a top-$k$ or threshold edge selection from the matrix $\tilde{A}^{(t)}$.

The final dynamic attention graph $\mathcal{G}$ is defined as the collection of the individual graphs $G^{(t)}$ at each timestep $t$ of token generation. It captures the token-to-token attention dynamics across the entire network and serves as the basis for our experimental analysis.

# 4 Methodology

## 4.1 Models, Data, and Evaluation

We analyse instruction-finetuned, state-of-the-art LLMs with model sizes spanning 1B–32B parameters. Our study includes the LLama family with sizes 1b and 8b, Mistral with 8b and 24b, Qwen with 1.5b, 3b, 7b, 14b, and 32b, and Gemma with 2b, 9b, and 27b

For data, we use the BigBench dataset (Srivastava et al., 2023) to extract tasks where emergence appears at this model size range. We identify prompts where models with 1B parameters usually fail, but models ranging from 8B–32B succeed. The chosen prompts encompass various domains, from basic logical reasoning to arithmetic and general knowledge tasks. Prompt examples can be seen in the appendix A.1.

To evaluate whether a model gets the answer correctly or not, we also utilize a similar approach to BigBench (Srivastava et al., 2023). We allow the model to answer with a single short sentence, a few words at maximum, and we simply match if the expected answer is in the given answer, essentially defining the problem as a simple classification problem. This can be done due to the simple nature of the prompts chosen and their objective responses, usually being a yes/no answer, a single word, or a number. However, we also note that our framework can be applied to analyse LLM behaviour on larger and more intricate questions, which may require reasoning and justification. To this end, we hypothesize that a good evaluation alternative for scaling these experiments would be to utilize LLM-as-a-judge (Zheng et al., 2023).

## 4.2 Analysis of Graph Features, Network Science Metrics

Upon generating the **Autoregressive Attention Graphs** for a given prompt and the generated answer, we then compute and analyse a series of graph-theoretic metrics and properties.

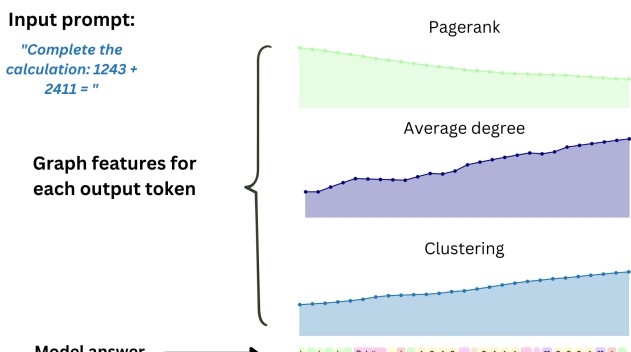

Figure 2: A visualization of features per output token. For each output token, we generate a separate graph from attention matrices and calculate selected features. Therefore, these graph features can be visualized as a time series over output tokens, with each token generated producing a new point for the time series.

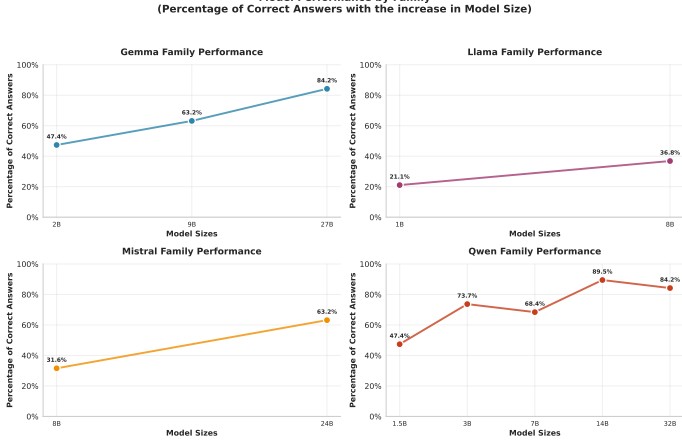

Figure 3: Model correctness on the given prompts. As model size increases, the percentage of successful tasks also increases, demonstrating increased correctness and emergent capabilities.

These graph features are computed for each snapshot of our evolving dynamic graph $\mathcal{G}$, and can then be visualized individually at each timestep of the generation process, or aggregated across timesteps for a global analysis of the dynamic graph. Figure 2 demonstrates how graph features are computed from a given prompt and answer.

After a comprehensive analysis of different graph features, we hand-picked three that have shown the most impactful results. They are as follows:

- Average Node Degree: The average node degree across all nodes of the graph. This can give us a general understanding of the inherent graph structure, such as its regularity.

- Maximum Pagerank: Representation of how uneven hub-spots are. This metric can indicate how specific tokens might be absorbing all attention or not. For some prompts, a higher max pagerank could potentially indicate that the model can ignore the irrelevant tokens better and attend to the actual question.

- Clustering Coefficient: The average clustering coefficient of the graph, which might provide insights into the capability of an LLM to generate and create attention substructures.

By applying these techniques, we aim to characterize how LLMs restructure their information flow with scale, potentially shedding light on the mechanisms behind emergent behaviour.

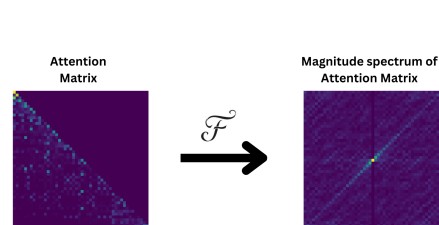

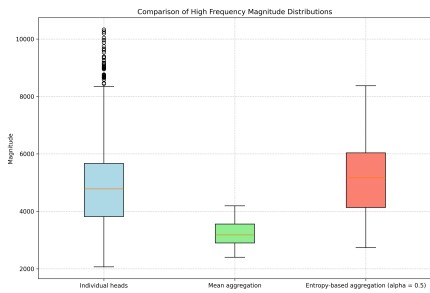

(a) An example of extracting magnitude spectrum from an attention matrix.

(b) The measure of high-frequency details for individual heads, the mean aggregation, and the entropy-based aggregation (alpha = 0.5)

Figure 4: Extraction of magnitude spectrum from the attention matrix. (4a) An example attention matrix and its corresponding Fourier magnitude spectrum. (4b) A comparison of high-frequency energy distributions across individual heads, mean aggregation, and entropy-based aggregation ($\alpha = 0.5$) demonstrates that the entropy-based method enhances high-frequency details.

## 5 RESULTS

### 5.1 PROMPT ANSWERS

Figure 3 shows the overall correctness of the various models to the given prompts, across model families and sizes. It can be inferred from the figure that increasing the model size improves the likelihood of the model answering correctly to a given prompt, demonstrating the emergent behaviour we wish to analyse.

### 5.2 A PRINCIPLED METHOD FOR HEAD AGGREGATION

The motivation for using entropy-based head aggregation was to capture the high-frequency details present in individual heads. The hypothesis was that a simple average across heads would act as a low-pass filter that discards the details in the attention. We can quantify the level of high-frequency details using a 2D Fourier transform. We first calculate the 2D spectrum of the attention matrix using the 2D Fourier transform, after which we calculate the magnitude of the spectrum. We calculate the high-frequency measure as the sum of the magnitude spectrum further than a fixed radius from the center. The selected threshold radius is half of the matrix dimension (in this case, the radius is half of the sequence length). Figure 4 illustrates this process and shows the results.

Figure 4a shows an example of an attention matrix and its corresponding magnitude spectrum. A main diagonal is interpreted as an edge between the upper-right and lower-left parts of the matrix, and therefore it appears in the magnitude spectrum as an edge on the opposite diagonal. Points closer to the center of the image represent lower frequencies, with the central point corresponding to the mean value of the original matrix.

Figure 4b shows the distribution of intensity of high-frequency components. It is shown for individual heads, for mean-aggregated heads, as well as for entropy-based aggregation of heads. We can see that mean aggregation significantly lowers the average intensity of high-frequency components. On the other hand, applying entropy-based aggregation with an alpha value of 0.5 (therefore applying median selection) retains the high-frequency intensity, even slightly shifting the distribution upwards.

### 5.3 FEATURE DISTRIBUTION BY MODEL SIZES AND FAMILIES

It is important to analyse whether some graph features show patterns inside model families, or with the overall increase of model size. Figure 5 shows how different features behave inside different families and with different model sizes. Figure 5a shows the distributions of features averaged over

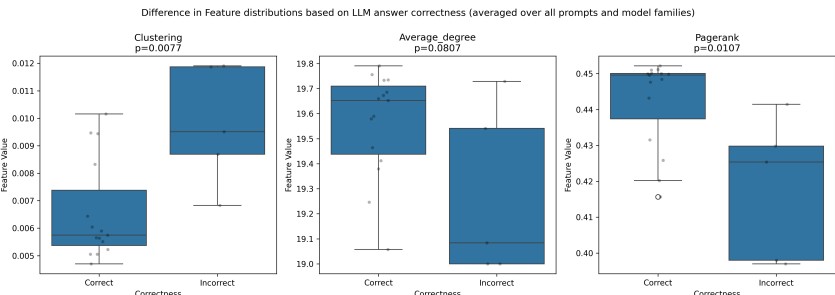

(a) Distribution of analysed features, based on their correctness for prompts, along with the measure of statistical significance. **Lower clustering coefficients and higher max pagerank values correlate with model correctness and emergent capabilities.**

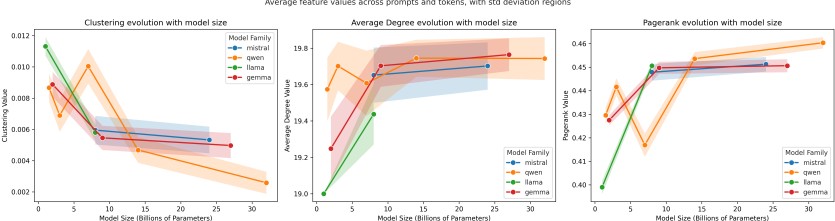

(b) Line plots showing the mean values of features over all the prompts and tokens, and their evolution with model size, for different model families. **Clear patterns between graph features and model size. As model size increases, both max pagerank and average node degree values increase, while clustering coefficients decrease.**

Figure 5: Graph features for different model families and model sizes averaged over all prompts, with the error bars showing the standard deviation between prompts. As model size increases, there are consistent patterns for clustering and pagerank. The only exception is the Qwen model family.

all model families and prompts, based on whether they answered the prompt correctly or not. In their titles, $p$-values are shown, as calculated using the Mann-Whitney U test. A value of $p$ lower than 0.05 indicates that a statistically significant difference exists between the distributions. As seen from the box plots in Figure 5a, clustering and pagerank features show statistical significance, while the average node degree feature does not.

Figure 5b shows the evolution of graph features over model sizes, averaged over prompts. Excluding the Qwen model family, all the models show a monotonic change as model size increases for all features.

# 6 DISCUSSION

## 6.1 ATTENTION PATTERNS ACROSS MODEL SIZE AND FAMILY

From our graph-theoretic analysis, we have found attention patterns across model sizes and families. As model size increases, all the model families except for the Qwen show clear trends for all features. Additionally, using the Mann-Whitney U test, we confirmed that there is a statistically significant difference in clustering and pagerank features for the correct and incorrect prompts. This further enhances the argument that the underlying structure of graphs formed from attention matrices impacts the ability of the models to answer correctly on a given prompt.

A possible explanation for the different behaviour of the Qwen model family is the difference in the pre-training data, as Qwen models had a significant part of the training data in the Chinese language, as claimed by the authors Bai et al. (2023).

## 6.2 QUANTIFICATION OF EMERGENT PROPERTIES VIA GRAPH METRICS

Our experiments reveal that while the average node degree does not provide a strong indication of emergent behaviour, both clustering coefficients and max pagerank metrics show a correlation to the correctness of responses and the emergent properties observed in larger models (see Figure 5a).

In particular, as model size increases and emergent behaviour becomes more pronounced, the clustering coefficients in the **Autoregressive Attention Graphs** trend towards lower values, 5b. This lower clustering suggests that tokens are less confined within tightly knit local clusters, allowing attention to be more globally distributed across the entire sequence. In other words, the model appears to better integrate long-range dependencies and capture a more holistic understanding of the context, thereby facilitating improved performance on complex tasks.

Similarly, the max pagerank values exhibit an upward trend with model scale, 5b. Higher pagerank scores indicate that certain tokens emerge as dominant hubs, absorbing a disproportionate amount of attention relative to others. This pattern is correlated with the correctness of the models and might imply that larger models are better at prioritizing critical tokens of the input while effectively filtering out irrelevant information from the prompt.

Thus, the increased unevenness in the attention distribution as quantified by max pagerank is quite interesting, as it showcases directly emergent capabilities related to internal understanding and mechanisms for emphasizing and prioritizing attention for key input elements, a skill which can be directly mapped to a critical ability of humans.

## 7    CONCLUSION

In this work, we introduced a novel interpretability framework that conceptualizes the attention matrices throughout autoregressive token generation as dynamic graphs. We crafted a novel technique for aggregating attention heads that is based on entropy. We show that this aggregation method is significantly better at capturing the high-frequency details, compared to the mean aggregation of heads.

With this framework, we then apply network science techniques to uncover if there exists a correlation between the patterns in attention graphs and the model's ability to answer correctly. Our analysis shows that there is a statistically significant difference in graph features between the models that can correctly answer the prompts and the ones that can not. Precisely, the difference is seen for clustering and pagerank features.

Furthermore, we have observed that in some cases, such as the Qwen model family, the attention dynamics exhibit different trends. We hypothesize that this difference appears due to the large differences in the pre-training data.

The biggest limitation of this work is the number of prompts used for analysis (due to computational and time constraints, only 3 prompts were used to provide analysis). Therefore, expanding the analysis to a larger number of prompts could potentially yield more informative conclusions. Specifically, it could be promising to explore how these graph properties may change across specific tasks, from arithmetic to reasoning and reading comprehension. However, we note that to effectively expand the analysis, it is important to establish an automated pipeline for verifying complex model outputs. We believe the approach used in (Srivastava et al., 2023) is overly simplistic, and we suggest the idea of using LLM-as-a-judge (Chen et al., 2024) for prompt answer correctness verification for future exploration.

Future work could also aim to integrate these findings into the training and fine-tuning process. For example, one promising direction is to identify attention graph features on larger models that correlate with correctness, and then rewire the attentions of a smaller model to mimic those patterns. Such an approach could clarify whether small models are inherently capable of achieving high performance if their attention is restructured, or if the knowledge itself is lacking.

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

## A  APPENDIX

### A.1  PROMPT EXAMPLES

The selected prompts cover a wide range of tasks including basic arithmetic, general knowledge, logical reasoning, reading comprehension, moral reasoning, fallacy detection, among others. Below are some examples of the prompts used:

> If all bloops are razzies and all razzies are lazzies, can we conclude that all bloops are lazzies?
>
> Calculate: 1256 + 2144 =
>
> What is the capital of France?
>
> Question: 120 is what percent of 50? Options: A) 5% B) 240% C) 50% D) 2% E) 500%
>
> At school today John was not present. He was away ill. All the rest of Ben's class were at school, though. When Ben got home after school his mother asked him, "Was everyone in your class at school today?" Ben answers, "Yes, Mummy". Did Ben's mom know that John was not at school?
>
> This AI is identifying whether statements contain fallacies. The AI responds with 'Valid' or 'Invalid' as appropriate. Statement: Excessive tanning is bad for the skin, so you should wear sunscreen to avoid it. Response:

