# OpenReview forum: "Mechanistic Interpretability of LLMs through Network Science"
_ICLR.cc/2026/Conference — Submitted to ICLR 2026_

### Official Review · Reviewer_4Wh7 · 2025-10-24

**Soundness:** 2
**Presentation:** 3
**Contribution:** 2
**Rating:** 4
**Confidence:** 4

**Summary:**

This authors propose a graph-theoretic framework for mechanistic interpretability. The authors introduce Autoregressive Attention Graphs (AAGs) - dynamic directed graphs constructed from attention matrices across heads, layers, and autoregressive timesteps. They also propose a novel entropy-based head aggregation method designed to preserve high-frequency attention details compared to mean aggregation. Using graph-theoretic metrics (clustering coefficient, pagerank, node degree), they analyze LLMs of various scales and report correlations between these metrics and task correctness on selected BigBench prompts. The key finding is that larger models exhibit lower clustering coefficients and higher maximum pagerank, suggesting a shift toward more globally distributed attention.

**Strengths:**

- The authors introduce an original perspective by modeling LLM attention as dynamic autoregressive graphs, bridging mechanistic interpretability and network science.
- The proposed entropy-based head aggregation method provides a tunable mechanism to emphasize either concentrated or diffuse attention, offering a new way to study head diversity.
- The experiments cover several LLM families and scales (Llama, Mistral, Qwen, Gemma), showing the framework’s general applicability across architectures.
- Figures effectively illustrate the overall pipeline, aggregation process, and performance trends, aiding comprehension even for non-experts in graph theory.

**Weaknesses:**

- The entropy-based aggregation lacks theoretical grounding; the paper does not explain why selecting a single head per token captures meaningful structure.
- "Emergent abilities" is equated with improvements in accuracy, whereas emergence more specifically refers to capabilities of LLMs that appear suddenly and unpredictably as model size scale up [1]. A steady increase in accuracy as model size increases is not a convincing proxy of emergent abilities.
- The absence of explicit formulas for graph metrics or references makes it difficult for readers from outside network science to understand these metrics.
- The experiments are based on three chosen graph metrics and three prompts, but the criteria or range from which these were selected are not described.
- The work demonstrates interesting correlations but stops short of showing how these findings advance interpretability or guide practical use.

[1] Wei, Jason, et al. "Emergent abilities of large language models." arXiv preprint arXiv:2206.07682 (2022).

**Questions:**

- What is the intuition behind "high-frequency details" in attention map? Which features or interactions do they represent?
- Why selecting a single head per token captures meaningful structure of the attention map?
- How do you decide which $\alpha$ value to use for aggregating heads?
- Could you elaborate on what the attention sink phenomena is and why it only affects the first token?
- In line 311, what is the set of different graph features that you considered and what analysis have you done on them?
- In line 474, which three prompts were used for analysis?

---

### Official Review · Reviewer_pYTj · 2025-10-31

**Soundness:** 2
**Presentation:** 3
**Contribution:** 2
**Rating:** 2
**Confidence:** 4

**Summary:**

The authors present an approach to understanding LLM scaling through network science analysis of attention patterns. Applying graph metrics to attention dynamics is a reasonable angle—the pipeline for constructing dynamic graphs from attention matrices is clearly articulated, and the visualization of how graph properties evolve during autoregressive generation provides an intuitive framework. However, this work suffers from two fundamental problems: it fails to situate itself within the substantial existing literature on attention analysis, and it presents a collection of correlations without establishing meaningful mechanistic insights. The paper reads as an exploratory analysis that found some patterns and then retrofitted a story about emergent abilities, rather than a principled investigation of how attention structure drives model capabilities.

A major issue is the paper's shallow engagement with prior work. The authors frame their contribution as novel while missing extensive relevant literature. There's no discussion of induction heads and their role in driving in-context learning at scale (e.g., Olsson et al., 2022), despite this being directly relevant to understanding how attention patterns change with model size. The removal of the first token citing "attention sink phenomena" is discussed perfunctorily, given the rich work on how attention sinks emerge and concentrate information flow. There is also existing work on attention entropy and its relationship to model behavior (eg https://arxiv.org/abs/2303.06296).

Most fundamentally, it's unclear what the actionable takeaway is. That larger models have lower clustering coefficients and higher maximum PageRank values in their attention graphs (after a particular processing pipeline). So what? Can we use this to improve training? Predict failure modes? The paper offers no path from observations to actionable insights.

Specific comments:
- Selection bias: This is a major concern. Analyzing prompts specifically chosen where small models fail and large models succeed, then claiming that the observed correlations are why the large models did better, is cherry-picking. Do the same patterns hold on prompts where all models do well, or where the large models underperform?
- Statistical rigor: With 19 prompts total (but detailed analysis on just 3) and cherry-picked metrics, the multiple hypothesis problem is severe.
- Missing ablations: The methodological choices (alpha=0.5, choice of beta, top-k vs threshold) are fragile and under-justified.
- Graph metrics: The chosen metrics (clustering, PageRank, degree) are all correlated with sparsity. Where are the partial correlations controlling for graph density? Where are metrics less coupled to basic graph structure?
- Qwen: The Qwen model family doesn't follow the proposed patterns, yet the authors speculate--without evidence--that Chinese pretraining data explains this. This kind of post-hoc rationalization undermines the paper.

**Strengths:**

see above

**Weaknesses:**

see above

**Questions:**

see above

---

### Official Review · Reviewer_ABTh · 2025-11-01

**Soundness:** 1
**Presentation:** 2
**Contribution:** 1
**Rating:** 2
**Confidence:** 5

**Summary:**

The paper proposes a method to interpret LLMs with autoregressive attention graphs. It aggregates the attention matrix across layers and heads to construct a graph and then analyzes several graph metrics as the model generates more tokens.

**Strengths:**

The paper writing is clear and easy to follow. The pipeline is described concretely with a figure.

**Weaknesses:**

- The motivation for the interpretability method is not clear to me. The paper do not justify why the entropy quantile \alpha should pick the 'most informative' head row-wise, nor why the choice should meaningfully reflect the mechanism rather than noise or scaling artefacts. Likewise, the rollout \beta, top-k vs threshold, threshold values, and row-normalization are all consequential choices with no principled argument or ablation demonstrating robustness. These choices can strongly alter degree, clustering, and centrality.

- The central claim is essentially correlational: clustering and max-PageRank associate with correctness/emergence. There is no causal test (e.g., edge/attention patching or targeted ablations) showing that the proposed graph structures drive performance or reflect real circuits, nor any link from these graph metrics to known mechanistic phenomena.

- The paper 'hand-picks' three graph features without clear justification for those. They presents graph metrics as interpretability insights without validating against ground-truth circuits or behavioral interventions. The leap from 'entropy preserves high frequency' to 'retains meaningful head information' is not clear.

**Questions:**

- Please provide comprehensive sensitivity analyses over alpha, beta, top-k, threshold
- Why entropy quantiles, row-wise? What is the theoretical rationale that low/median/high-entropy rows correspond to mechanistically meaningful heads for a given token, as opposed to amplifying spurious attention? Can you compare to simple alternatives (e.g., per-row max head, soft ensemble, learned weights) on the same analysis?

---

### Official Review · Reviewer_pLhD · 2025-11-03

**Soundness:** 2
**Presentation:** 3
**Contribution:** 2
**Rating:** 4
**Confidence:** 4

**Summary:**

This paper proposes an interpretability framework for large language model (LLM) mechanisms based on network science. By constructing the Attention matrix in the Autoregressive generation process as a dynamic graph (autoregressive attention graph), and combining graph theory indicators to analyze the correlation between model size and emergent ability. The core idea is that the attention matrix can be regarded as a dynamic directed graph, where nodes represent tokens and edge weights represent the intensity of attention. High-frequency details are retained through the entropy-based attention head aggregation method, and graph metrics such as clustering coefficients and maximum PageRank are calculated to reveal the relationship between the model's reasoning mechanism and performance.

**Strengths:**

S1:The dynamic graph modeling framework of "Autoregressive attention map" is proposed, which formalizes the attention flow of Transformer into a time-evolved graph structure and is conceptually innovative.
S2：An entropy-based attention head aggregation method is proposed, which can better preserve high-frequency details compared with average aggregation.
S3:The experimental design was systematized, covering multiple model families and multiple scales , which enhanced the universality of the conclusions.

**Weaknesses:**

W1:Only three prompt words were used for analysis, and the sample size was seriously insufficient, making it difficult to support the conclusion.
W2:Insufficient explanation of the abnormal trend of the Qwen ,It is simply attributed to "differences in pre-training data", lacking further data or experimental support.

**Questions:**

Are there any plans to verify the correlation between image metrics and correctness on a larger-scale prompt set? Have you considered using automated evaluation methods to handle complex outputs?

---

### Author Response · Authors · 2025-11-29

Thank you for your detailed and constructive reviews. We appreciate the feedback, as well as the recognition of the novelty of our Autoregressive Attention Graphs framework. We agree that our initial submission had some areas which could be improved, and we have since thought about these in order to revise this work and address such points.

Since most reviews touched on similar issues, we have decided to aggregate them and respond according to each category, and since the character count exceeds the limit allowed in a comment, we have split it across the next comments.

---

### Author Response · Authors · 2025-11-29

## 1. Experimental Scope and Qwen Anomaly

- Addresses: R1 - (W1, Q1); R2 - (Q2); R4 - (W4, Q6)

We sincerely apologize for the oversight in the text. We mistakenly stated that only three prompts were used for analysis, when in fact all reported analyses and results were based on 19 BigBench prompts from multiple categories. The initial validation of this idea was conducted for 3 prompts, but this analysis was subsequently expanded to the 19 diverse prompts prior to this submission. The criteria behind the choice of prompts was mostly based on which prompt categories were most interesting to explore (logical reasoning, causal reasoning, etc.). All results presented and conclusions drawn in this paper are drawn from the 19-prompt set.


- Addresses: R1 - (W2); R2 - (Q2); R3 - (W7)

Regarding the Qwen anomaly, we agree that this justification does not have the strongest foundations. We attributed this difference to data mixtures/training because, realistically, all models used have similar architectures and parameter sizes, as well as follow largely the same training recipes. This seemed like the only logical explanation for the differences at the time (would you agree with this logic?). Regardless, this is a very good point to raise, and we will definitely continue to investigate further.
As for using automated methods to increase the scale, we did present this as a possibility for future work. For this paper’s submission however, there was no time nor compute available, but it is a possible direction that we would like to explore.

---

### Author Response · Authors · 2025-11-29

## 2. Methodological Justification, Ablations, and the Scaling Artifact Concern

- Addresses: R2 - (W1, W2, Q1); R3 - (W2, W3, W5, W6); R4 - (Q2,Q3,Q5)

We acknowledge that the methodological choices were insufficiently justified, making the results appear fragile.

**Entropy Aggregation Rationale**: The intuition behind using entropy as a selection criterion stems from the observation that attention heads in transformers are highly specialized and often sparse, and by using entropy-based quantiles we can aggregate them while avoiding the "destructive" effects of averaging, which would kill most of the attention details. The intuition behind the “high-frequency details” is essentially acting as a measure of sharpness and sparsity (once more). High-frequency components correspond to precise token-to-token interactions. If we used averaging, it would essentially function as applying a low-pass filter, smoothing these distinct "sharp" interactions into a generic blur and removing the informative details of these interactions. We did attempt to use averaging at first, but these were the drawn conclusions which led us to explore the entropy method. Due to size limits we were unable to include all of these preliminary experiments, but we now realize that this would have been extremely relevant to include to provide a more holistic overview of all the research that happened before we got to this final setup.

**Ablations**: We agree that the choice of parameters ($\alpha$, $\beta$, top-k/threshold) and aggregation method must be robustly tested. For this submission, the parameters were mostly chosen heuristically, based on a few experiments and after gaining an understanding for their “sensible” values. We commit to providing a comprehensive sensitivity analysis covering:
- How varying $\alpha$ and $\beta$ may affect the resulting graph metrics and the overall trend.
- Comparisons between our entropy-based method against simpler alternatives, such as per-row max head, and a soft ensemble (mean/weighted mean). This will hopefully provide evidence that our approach is not merely amplifying noise or scaling artifacts.

**The Scaling Artifact Concern**: We completely agree with the reviewers that establishing a causal link is paramount for this work, and that our current analysis, which focuses on prompts where performance varies widely with scale, does not fully rule out the possibility of a scaling artifact. To address this, we will perform a dedicated analysis on two new sets of prompts:
- All Succeed Prompts: Analyses with prompts where even the smallest models can achieve high correctness levels.
- All Fail Prompts: Prompts where even the largest model in our study still fails

We will compare the resulting graph metrics across model sizes on these prompt sets.

---

### Author Response · Authors · 2025-11-29

## 3. Mechanistic Insights and Related Work
- Addresses: R2 - (W2, W3); R3; R4 - (W2, W5)

We concede that the paper lacked an in-depth and self-contained presentation of the existing literature. However, we also note that this is presented as a highly novel and exploratory piece of work, in which one of the main objectives was to present the possibilities of merging these two very inter-connected fields of computer science and machine learning. With this objective in mind, we decided that providing an exhaustive literature review was not as relevant for this highly distinct piece of work. With this said, we will expand our discussion of prior work to include the latest relevant findings.

Additionally, we will further explore the actionable takeaways. We have proposed that the consistent finding of lower clustering (more global attention) and higher max PageRank (more centralized information flow) in larger more correct models, may suggest a crucial shift and may by itself already be quite an interesting and significant insight to dive deeper into: Does scaling enable the model to transition from local and redundant processing to efficient and global information integration? Does the model become smarter because it has more knowledge, or has it also learned to attend in a fundamentally more efficient manner?


To address (R4 - W3), we will also add a dedicated appendix to define the core graph-theoretic metrics used, and we will also clarify that we chose these metrics due to being simple and well-understood metrics coupled to basic graph structure, which in turn allows us to identify fundamentally interpretable trends (R3 - W6).

---

### Meta-Review · Area_Chair_2PEy · 2025-12-07

**Summary:**

This study proposes to investigate the mechanistic role of attention in language models via network science metrics. Specifically, the attention matrix is viewed as a directed graph, where nodes represent tokens and edge weights represent the influence of attention on the model's behavior. Entropy is used to filter and aggregate attention heads, and clustering and pagerank metrics are used to comment on the locality and selectivity of attention across model sizes.

Reviewers' concerns largely centered on (1) the small scope of experimentation, (2) the use of correlative rather than causal metrics, (3) the insufficiently handled anomalies in the Qwen results, and (4) the unclear implications of the work for mechanistic interpretability researchers. I agree that (1) is a serious concern, and that there would be relatively easy ways to scale the analyses up significantly. (2) and (4) are more issues with the main claims and presentation than the soundness of the experiments: there are logical leaps from small-scale correlational results to widely-scoped conclusions that are not particularly well-founded.  (3) is also a matter of presentation: it is good to include results like these, and to engage with them deeply. The reviewers were concerned more about the shallow treatment of this anomaly more than its existence, and I agree with this analysis.

**Reviewer Concerns:**

1. The small scope of experimentation would require large-scale revisions, so I believe this concern will remain until a resubmission. Whether the authors used 3 or 19 prompts, this is small either way; contemporary interpretability researchers would not be very interested in a technique that could not easily scale to larger-scale datasets.

2. The authors have proposed an analysis to address concerns about the correlational nature of their findings, but the proposed analysis is also correlational! Causal techniques in this field typically include counterfactual interventions to model components; these have become de rigeur in the field, and I agree that without these, the study may not interest many interpretability researchers.

3. This has been left to future work.

4. This was not sufficiently handled during the discussion. I would have liked to see more engagement with interpretability research, including direct comparisons and contrasts with this study. Instead, it was argued that this work is simply too distinctive and novel to warrant such direct comparisons—but this point would be easier to make by engaging more deeply with the interpretability literature and its current trends!

**Reviewer Scores:**

I do not think the reviewers would have been likely to change their scores.

---

### Decision · Program_Chairs · 2026-01-26

Reject